# Effect of Red and Blue Light on the Growth and Antioxidant Activity of Alfalfa Sprouts

**Kelong Sun [1], Ying Peng [1], Mengyuan Wang [2], Weihu Li [3], Yang Li [3] and Jianjun Chen [2,*]**

1. College of Chemistry, Huazhong Agricultural University, Wuhan 430070, China; sunkelong@webmail.hzau.edu.cn (K.S.); miqian@webmail.hzau.edu.cn (Y.P.)
2. College of Engineering, Huazhong Agricultural University, Wuhan 430070, China; 2021310230227@webmail.hzau.edu.cn
3. Public Teaching Department, Tibet Agricultural and Animal Husbandry University, Nyingchi 860000, China; liweihu@xza.edu.cn (W.L.); liyang@xza.edu.cn (Y.L.)
* Correspondence: chenjianjun@mail.hzau.edu.cn

**Abstract:** Alfalfa sprouts are popular as a gourmet vegetable that contains a variety of antioxidants with anti-cancer and anti-coronary heart disease properties. In this study, under a photosynthetic photon flux density (PPFD) of 30 mol·L$^{-1}$ photoperiod of 12 h for 3 days, and a temperature of $25 \pm 2$ °C, we investigated the effects of different light qualities on the growth, nutritional quality and antioxidant activity of alfalfa sprouts by modulating LEDs with different red and blue ratios, and searched for suitable light-quality conditions for alfalfa sprout growth. The experimental results showed that the dark treatment favored the growth of alfalfa sprout hypocotyls and the increase of soluble sugar content; alfalfa fresh weight was the largest under the white and red light treatments; nitrate content was the lowest in the treatment with a red-to-blue ratio of 2:2 (2R2B); soluble proteins and total phenolic content were the highest in the treatment with red-to-blue ratio of 1:3 (1R3B); and the total antioxidant activity of sprouts was the highest in the blue light treatment.

**Keywords:** alfalfa sprouts; light quality; LED; nutritional quality; antioxidants

## 1. Introduction

Sprouts are seedlings that grow from the germination of seeds of various plants. Common sprouts include soybean sprouts, mung bean sprouts, alfalfa sprouts, radish sprouts and other sprouted vegetables. Sprouts are rich in protein, vitamins, fiber, minerals, polyphenols, flavonoids, and other beneficial ingredients, which have the benefits of lowering blood sugar, lowering blood lipids, promoting digestion, enhancing human immunity, and improving antioxidant capacity. Sprouts are full of nutrients, crisp and tasty, and loved by the public [1–4]. Sprouts have a short growth cycle and planting is not controlled by the season, only a small amount of water is needed every day, and there is no need to add the nutrient solution to meet the needs of its growth. Sprouts are suitable for growing in sheltered light or in low light, and can be eaten when they grow to 5–7 cm. Plant factories can produce vegetable crops on a large scale because they can provide a suitable environment for vegetable growth, make planting agriculture independent of seasonal control, and have advantages such as high planting space rate [5,6]. Moreover, a plant factory can not only adjust the light environment, $CO_2$ and suitable temperature needed by plants, but also can provide the nutrient solution needed by plants to quickly promote the growth of plants. Compared with traditional fluorescent lamps and high-temperature sodium lamps, LEDs have many advantages, such as adjustable light quality, energy saving, environmental protection, low radiation, small size, long service life, etc., and have been widely put into use in plant factories [7–9].

As an extremely important environmental factor in plant growth, light plays a crucial role in regulating plant growth and development, promoting photosynthesis, and even

promoting the expression of some genes [10–15]. Photosensitive pigments, cryptochromes and other receptor proteins related to photosynthesis exist on the surface of plant cells. Among them, photosensitive pigments mainly absorb red and far-infrared light, and cryptochromes mainly absorb blue and near-ultraviolet light [16,17]. Previous experimental results have shown that red light promotes morphological indicators such as stem length, leaf area, and fresh weight, while blue light promotes the opening of stomata and the synthesis of photosynthetic pigment proteins in plants [18–21]. There have also been a number of studies on LED light quality to enhance the quality of sprouts. The experimental results of Qian et al. showed that blue light significantly enhanced the total phenolics, anthocyanin content and antioxidant levels, palatability and nutritional value of Chinese kale sprouts [22]. Ruiz-Nieto et al. concluded that green light increased lentil sprout stem thickness, violet and white lights increased soluble protein content, and blue light increased seed germination, β-carotene and phenolic compound concentrations, and antioxidant activity [23].

Alfalfa sprouts, as one of the sprouted vegetables, have cholesterol-lowering, anti-diabetic and anti-obesity properties [3]. To date, there have been a number of studies on alfalfa sprouts. Fiutak et al. showed that RGB-light-treated alfalfa sprouts were more effective and had higher β-carotene, lutein, and polyphenol contents than cool white light and warm white light [24]. The findings of Zhang et al. illustrated that more isoquercitrin was produced in soybean sprouts under dark conditions, while more isoflavones were produced in mung bean sprouts after light exposure [25]. The effects of red and blue light, as well as mixed red and blue light, on the nutritional quality and antioxidant activity of alfalfa sprouts have been little studied. So it is necessary for improving the quality of alfalfa. Here, we used different ratios of red and blue LEDs as growth light sources for alfalfa sprouts to investigate the effects of different red and blue lights on the growth of hydroponically grown alfalfa sprouts, the content of soluble sugars, soluble proteins, nitrate and antioxidant properties, aiming at providing alfalfa sprouts with a spectral scheme of alfalfa sprouts for growth in the plant factory.

## 2. Materials and Methods

### 2.1. Materials and LED Equipment

Alfalfa (*Medicago sativa* L.) seeds, variety Blue Moon, were purchased from the Xintai Zhouquan Agricultural Technology Co., Ltd., Xintai, China. BSA and gallic acid were purchased from Aladdin, Shanghai, China. Others reagents were purchased from Sinopharm chemical reagent Co., Ltd., Shanghai, China. All reagents and drugs were of analytical purity. The spectral data measurement tool was a lighting passport purchased by Asensetek, Taibei, Taiwan. Alfalfa lighting environments were in ventilated black boxes equipped with LEDs (Shenzhen NN lighting Co., Ltd., Shenzhen, China). The LED light spectrum data are shown in Table 1.

### 2.2. Seed Germination and Light Treatments

Seeds of 8.0 g of alfalfa were weighed and soaked inside warm water at 40 °C for 30 min, and then were soaked for 12 h at room temperature. The soaked seeds were picked up, transferred to seedling trays, and placed in a shady and ventilated place to germinate, and water was sprinkled on the surface once a day. When the alfalfa sprouts grew to around 3 cm, they were placed under different red and blue light to cultivate and grow, with a light intensity of 30 $\mu mol \cdot m^{-2} \cdot s^{-1}$, a light photoperiod of 12 h, and a temperature of 25 ± 2 °C. Here, we set up six different experimental treatments: red (R), blue (B), a red-to-blue ratio of 3:1 (3R1B), a red-to-blue ratio of 2:2(2R2B), a red-to-blue ratio of 1:3 (1R3B), white (W) and one control treatment dark (D). After 3 days of light exposure, a number of alfalfa sprouts were randomly selected to investigate the effects of different red and blue light on the growth and antioxidant activity produced by alfalfa sprouts.

**Table 1.** Main spectral parameters of LED lights for cultivating alfalfa sprouts.

| Light Quality | Light Quality Ratio | Color Temperature (K) | CIE1976 | λmax (nm) |
|---|---|---|---|---|
| Dark | — | — | — | — |
| Red | 100%Red | — | $u' = 0.5552$, $v' = 0.5153$ | 660 |
| 3R1B | 75%Red25%Blue | — | $u' = 0.2945$, $v' = 0.4170$ | 450 660 |
| 2R2B | 50%Red50%Blue | — | $u' = 0.2605$, $v' = 0.1634$ | 450 660 |
| 1R3B | 25%Red75%Blue | — | $u' = 0.2209$, $v' = 0.1198$ | 450 660 |
| Blue | 100%Blue | — | $u' = 0.1975$, $v' = 0.0869$ | 450 |
| White | 100%White | 2385 | $u' = 0.2945$, $v' = 0.4170$ | 450 620 |

*2.3. Determination of Hypocotyl Length*

Hypocotyl length: alfalfa sprouts were taken from each treatment, roots were removed, and the hypocotyl length of alfalfa was measured using a digital vernier caliper.

*2.4. Determination of Fresh Weight Content*

Ten alfalfa sprouts were taken from each treatment, the attached seed coat was removed, and the fresh weight of each 10 sprouts was measured.

*2.5. Determination of Dry Weight Content*

The alfalfa after fresh weight measurement was placed in an oven at 105 °C for 30 min, cooled down to 80 °C for drying to a constant weight, and the dry weight of the alfalfa sprouts was determined.

*2.6. Determination of Moisture Content*

The moisture content of alfalfa = (1 − dry weight/fresh weight) × 100%.

*2.7. Determination of Physiological Parameters*

2.7.1. Soluble Sugar

Soluble sugars were determined using the anthronesulfuric acid colorimetry method [26]. We weighed 1.0 g of alfalfa sprouts and mixed them with 10 mL of distilled water extraction, in a boiling water bath for 20 min. The mixed solution was filtered through a 0.22 μm filter head, and 0.5 mL of the sample solution was taken. A total of 5 mL of concentrated sulfuric acid, 0.5 mL of anthrone ethyl acetate solution, and 1.5 mL of water were added and shaken well. The mixed solution was boiled in a water bath for 12 min, cooled to room temperature and then the absorbance value was measured at 620 nm (0–100 mg·L$^{-1}$ sucrose as a standard solution of soluble sugars).

2.7.2. Soluble Protein

Soluble protein content was determined using Coomassie brilliant blue G-250 reagent [27]. We weighed 1.0 g of alfalfa sprouts, ground them and added 10 mL of distilled water for extraction, and centrifuged it at 4000 r/min for 20 min at 4 °C. We filtered it with a 0.22 μm filter head, aspirated 1 mL of supernatant, added 5 mL of Coomassie Brilliant Blue G-250, and mixed it and let it stand for 2 min, and absorbance values were measured at 595 nm (0–100 mg·L$^{-1}$ BSA as a standard solution for soluble proteins).

### 2.7.3. Nitrate

Nitrate content was determined by a colorimetric method using salicylic acid [27]. We weighed 1.0 g of alfalfa sprouts into a 20 mL graduated test tube, added 10 mL of deionized water, placed it into a boiling water bath for 30 min, cooled and filtered it, and then condensed it into a 25 mL volumetric flask. We took 0.1 mL of test solution, added 0.4 mL of 5% salicylic acid–sulfuric acid solution. After standing for 30 min, 9.5 mL 8% NaOH solution was added, it was cooled to room temperature, and we measured the absorbance value at 410 nm ($0$–$100$ mmol·$L^{-1}$ potassium nitrate as a standard solution for nitrates).

### 2.7.4. Total Phenolic Compounds

Total phenolic content was determined using Folin–Ciocalteu's reagent [28]. We weighed 0.5 g of alfalfa sprouts, grinded them with the assistance of liquid nitrogen, extracted it with 10 mL of finishing water in a centrifuge tube, centrifuged it at 4000 r/min for 20 min at 4 °C, and filtered it with a 0.22 μm filter head. A total of 1 mL of the supernatant was taken in a test tube, then 1 mL of 1.0 M Folin–Ciocalteu's reagent and 3 mL of 10% $Na_2CO_3$ were added. The absorbance value was measured at 765 nm after incubation at 30 °C for 30 min ($0$–$100$ mg·$L^{-1}$ Gallic acid as a standard solution for total phenolic content).

### 2.7.5. Total Antioxidant Activity

Total antioxidant activity was determined by the DPPH method [10]. We weighed 0.008 g of DPPH and dissolved it in anhydrous ethanol to configure a $2 \times 10^{-4}$ mol·$L^{-1}$ solution of DPPH. We weighed 1.0 g of alfalfa sprouts, used liquid nitrogen to assist grinding, extracted it with 10 mL of anhydrous ethanol in a centrifuge tube, and filtered it with a 0.22 μm filter head. We took 3 mL of DPPH solution, added 3 mL of sample solution, mixed well, and measured the absorbance at 518 nm, which was recorded as Ai. We took 3 mL of sample solution, added 3 mL of anhydrous ethanol, mixed it well in the dark, and measured the absorbance at 518 nm, which was recorded as Aj. We then took another 3 mL of DPPH solution, added 3 mL of anhydrous ethanol, mixed it well in dark, and measured the absorbance at 518 nm, which was recorded as A0.

$$\text{DPPH radical scavenging rate (\%)} = (1 - Ai - Aj)/A0 \times 100\%$$

### 2.8. Statistical Analyses

The hypocotyl length of alfalfa sprouts was tested in 30 independent replicates, fresh and dry weights in 5 independent replicates, and biochemical indices in 3 independent replicates. Data processing was performed using SPSS 26. Tukey's LSD was used to test data for significant differences and Duncan's practical for post hoc comparisons of differences between multiple treatments ($p \leq 0.05$ indicates a significant difference).

## 3. Results and Discussion

### 3.1. Morphology of Alfalfa Sprouts Treated with Different Light Qualities

Figure 1 illustrates the growth of alfalfa sprouts after 3 days of light exposure to different red and blue lights. At a cursory glance, the difference between sprouts from the dark and light treatments is quite significant. The cotyledons of sprouts in the dark treatment had a yellow color, while the cotyledons of sprouts in the light treatments had a soft green color. The hypocotyl length of alfalfa sprouts under different red and blue light illumination does not have significant difference from that under white light illumination.

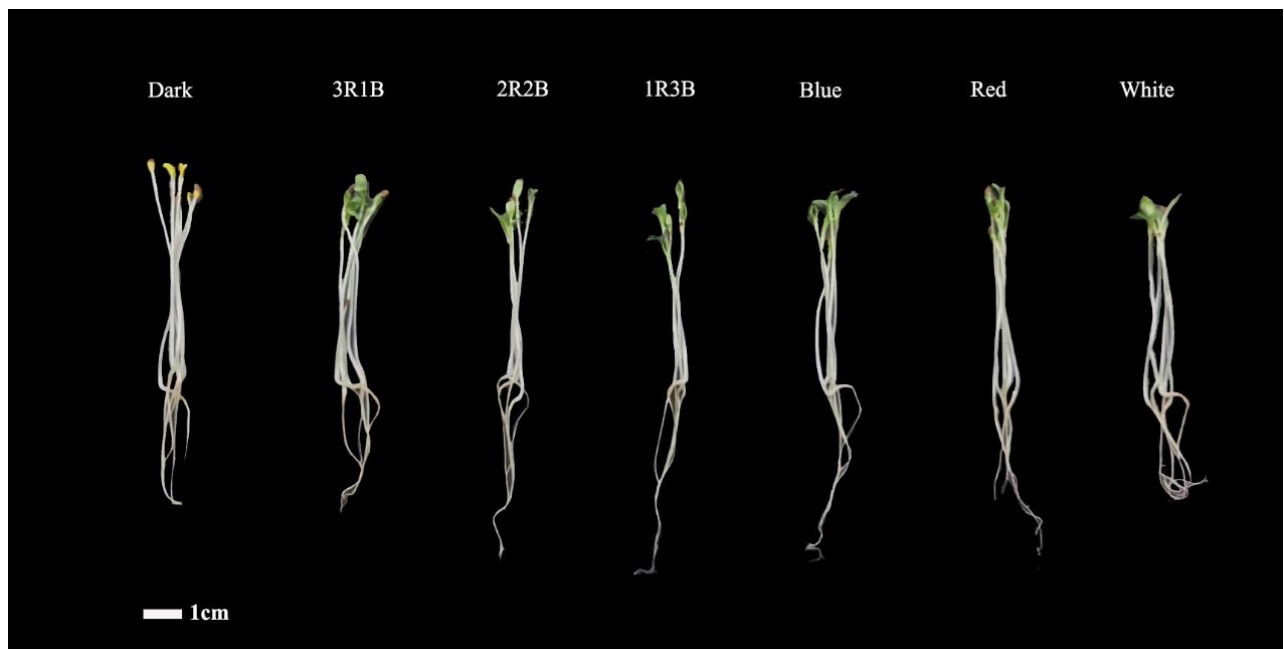

**Figure 1.** Morphology of alfalfa sprouts by different light qualities after 3 days of light exposure.

*3.2. Effect of Different Light-Quality Treatments on the Fresh Weight, Dry Weight, and Water Content of Alfalfa Sprouts*

Light has a significant impact on plant growth and morphology, and different plants require different light qualities, light intensities, and photoperiods. Table 2 shows the effects of different light qualities on the hypocotyl length, fresh weight, dry weight and moisture content of alfalfa sprouts. As can be seen from Table 2, alfalfa sprouts in the dark treatment had the longest hypocotyl length of 45.50 mm, which was 21.18–31.72% more compared to the other treatments, and there was a highly significant difference with the other treatments. Except for the dark and red light treatments, there was little difference in the effect of light treatments on the hypocotyl length of alfalfa in the rest of the treatments. The experimental results indicated that treatment under dark conditions was more favorable for the increase in hypocotyl length in alfalfa sprouts. Lorena experimental results also showed that the hypocotyls of carrot sprouts under dark conditions were much higher than those of the other light treatments, and there was little difference in the hypocotyls of the other light treatments [29]. Although red light promoted the growth of hypocotyls to a certain extent, the effect was not obvious, probably due to the low light intensity (30 μmol·m$^{-2}$·s$^{-1}$) and light duration of only three days [30].

**Table 2.** Effect of different light qualities on the hypocotyl length, fresh weight, dry weight and the moisture content of alfalfa sprouts.

| Light Quality | Hypocotyl Length (mm) | Fresh Weight (g) | Dry Weight (g) | Moisture Content (%) |
|---|---|---|---|---|
| Dark | 45.5035 ± 5.1324 a | 0.1968 ± 0.0203 c | 0.0104 ± 0.0013 a | 94.73 ± 0.2873 e |
| Red | 37.5497 ± 3.4732 b | 0.2184 ± 0.0102 ab | 0.0099 ± 0.0009 a | 95.46 ± 0.2223 a |
| 3R1B | 36.9157 ± 3.8812 bc | 0.2130 ± 0.0161 abc | 0.0098 ± 0.0012 a | 95.41 ± 0.2045 ab |
| 2R2B | 34.6500 ± 3.1438 d | 0.2156 ± 0.0192 abc | 0.0105 ± 0.0009 a | 95.13 ± 0.2187 bcd |
| 1R3B | 34.5455 ± 3.5863 d | 0.1958 ± 0.0133 c | 0.0096 ± 0.00204 a | 95.10 ± 0.1352 cd |
| Blue | 35.1050 ± 3.5105 cd | 0.1986 ± 0.0075 bc | 0.0101 ± 0.0002 a | 94.90 ± 0.2394 de |
| White | 35.6890 ± 4.1732 bcd | 0.2202 ± 0.0100 a | 0.0102 ± 0.0006 a | 95.37 ± 0.1635 abc |

Data are shown as the mean ± S.E. Values followed by different lowercase letters within each column are significantly different at $p \leq 0.05$. The same as below.

The fresh weight of alfalfa sprouts in the white light treatment was the greatest, but it was not significantly different from the red light, 3R1B, and 2R2B treatments, which was only 2.51–3.38% more, whereas it was 10.88–12.46% more compared to the dark, 1R3B, and blue light treatments, which was significantly different ($p < 0.05$). The results of the experiment showed that the increase in the proportion of red light did favor the increase of alfalfa hypocotyls and fresh weight to a certain extent, which was similar to the experiments done by the former [13].

The dry weight of all treatments were not significantly different ($p > 0.05$), and the largest dry weight of treatment 2R2B was only 9.37% more than that of the smallest treatment 1R3B. It indicates that the light and dark treatments had similar effects on organic matter accumulation in alfalfa sprouts, but the composition of organic matter needs to be further explored.

The moisture content of sprouts reflects to some extent the freshness and flavor of sprouts [31]. The highest moisture content of alfalfa sprouts was 95.45% in the red light treatment, followed by the white light and 3R1B treatments, and the lowest moisture content of alfalfa sprouts in the dark and blue light treatments.

### 3.3. Effect of Different Light Quality Treatments on the Soluble Sugar Content of Alfalfa Sprouts

Sugars are not only the main source of energy and structural components of the plant body, but also important regulators that promote gene expression [32]. Sugars, as intermediate or final products produced by metabolic processes, not only regulate physiological processes such as plant growth and development, osmosis, and resistance, but are also involved in intracellular signaling regulation and conduction [33]. In general, different ratios of red and blue light have different effects on plant soluble sugar content. Dong et al. showed that the soluble sugar content of tomato fruits in the 3R1B light treatment was higher than the other treatments [34]. Zhang et al. showed that the 8R1B treatment significantly promoted the growth, biomass accumulation, and soluble sugar content of Chinese Kale [35]. Figure 2 shows the effects of different red and blue light on the soluble sugar content of alfalfa sprouts. Among them, alfalfa sprouts in the dark treatment had the highest soluble sugar content of 2.60 mg/g, which was 67.38–103.72% more than the other treatments. The soluble sugar content of sprouts in the red light treatment was the next highest at 1.56 mg/g, and the soluble sugar content of alfalfa sprouts in the white light treatment was the lowest at 1.28 mg/g. The results of this experiment indicate that dark treatment favors the increase of soluble sugar content in alfalfa sprouts, which is similar to the findings of Linda et al. [36].

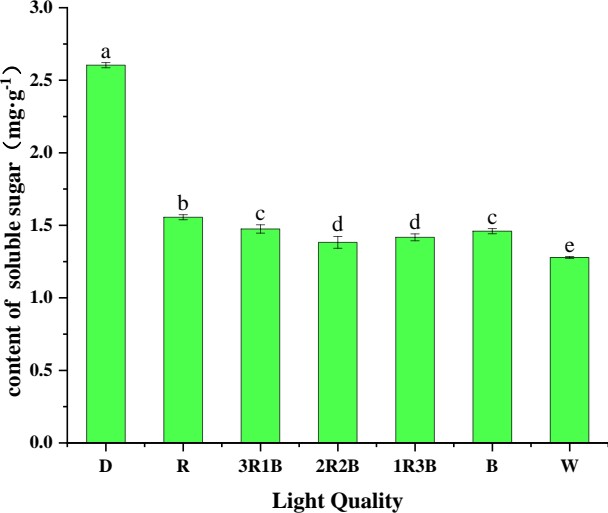

**Figure 2.** Effect of different light qualities on soluble sugar content of alfalfa sprouts (abbreviation: D, dark; R, red; B, blue; W, white. The same as below).

*3.4. Effect of Different Light-Quality Treatments on the Soluble Protein Content of Alfalfa Sprouts*

Soluble protein is an important physiological and biochemical index in plants, providing plant nutrients and participating in the osmotic regulation and metabolism of plants. Soluble proteins include a number of enzymes involved in metabolism, and their content can be used as an indicator of the total metabolic capacity of the plant body. Figure 3 shows the effect of different red and blue light on soluble protein content of alfalfa sprouts. Among them, the 1R3B treatment had the highest soluble protein content of 1.56 mg/g after light exposure, which was 8.27–27.52% more compared to the other treatments. 2R2B treatment had the second highest soluble protein content, and the dark treatment had the lowest soluble protein content. Studies have found that the red to blue ratio of 6:3 favors the increase of soluble protein content and the accumulation of antioxidant substances such as anthocyanins and flavonoids in Chinese Kale [35]. Other studies concluded that 2:1 mixed red and blue color light was beneficial to the soluble protein content and antioxidant activity of black-eyed pea sprouts [37]. This experiment shows that light treatments are beneficial to the increase in soluble protein content, and that red and blue compound light is to some extent more effective than single red light or single blue light.

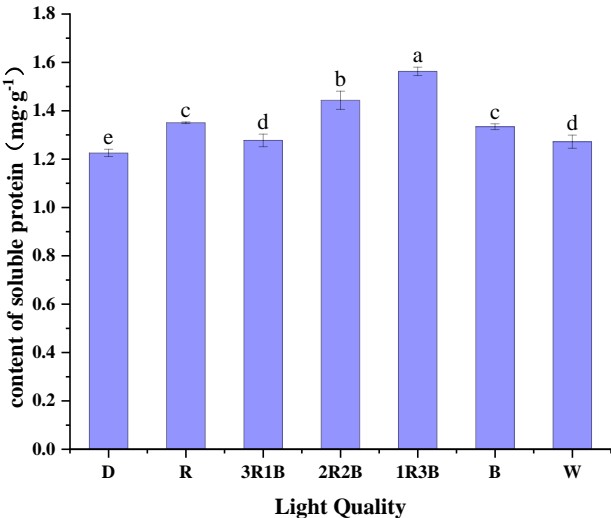

**Figure 3.** Effect of different light qualities on soluble protein content of alfalfa sprouts.

*3.5. Effect of Different Light-Quality Treatments on Nitrate Content of Alfalfa Sprouts*

Nitrate is an important form of N and is easily absorbed by the plant body. Nitrate itself is harmless, but if it is converted to nitrite by other means in the plant body and consumed by humans, it will bring a risk of cancer [38,39]. So reducing nitrate levels in plants via light quality is essential. Figure 4 shows the effect of different red and blue light treatments on nitrate content of alfalfa sprouts. The white light treatment had the highest nitrate content of 342.51 mg/g, which was 23.87–68.95% more compared to the other treatments, and had highly significant differences with the other treatments. This was followed by the dark treatment with 267.51 mg/g of nitrate. The lowest content of nitrate in alfalfa sprouts was 202.73 mg/g in the 2R2B treatment. Zhou et al. showed that mixed red and blue light was more effective than monochromatic red light in reducing nitrate content of lettuce, and the effect was more obvious in the R/B = 4 treatment [38]. He et al. showed that red/white light = 2:1 can reduce the content of nitrate in cabbage leaves, and the increase in the proportion of red light nitrate reduction effect is not obvious [27]. Similar to the results of previous studies, the results of this experiment showed that red and blue compound color light could reduce the nitrate content of sprouts to a certain extent. Among them, the 2R2B treatment had the most obvious effect in reducing the nitrate content. The reason for this can be attributed to the fact that the mixed red and blue color light better promotes nitrogen metabolism and reduces nitrate accumulation in the plant.

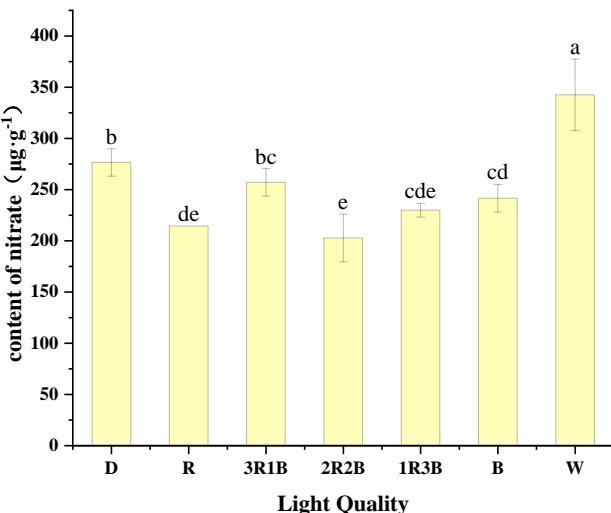

**Figure 4.** Effect of different light quality on nitrate content of alfalfa sprouts.

### 3.6. Effect of Different Light-Quality Treatments on the Total Phenolic Content of Alfalfa Sprouts

Phenolic compounds are widely found in plants and play an important role in antioxidant and free radical scavenging in plants, such as flavonoids, anthocyanins and other antioxidant substances [40–42]. In addition, phenolic compounds, which can confer the taste, aroma and color of vegetables and fruits, are an indispensable class of chemicals. Liu et al. showed that blue light significantly increased the content of all individual phenolics and increased pea sprout ABTS, FRAP, and reducing power. Grzegorz et al. showed that RGB-treated alfalfa sprouts had a higher total phenolic content than cool-white light and warm-white light, and both were higher than the dark treatment. Figure 5 shows the effect on total phenol content of alfalfa sprouts after different red and blue light treatments. After 3 days of light exposure, the highest total phenol content of alfalfa in the 1R3B treatment was 1.19 mg/g, which was 1.24–20.82% more compared to other treatments. Alfalfa sprouts in the dark treatment had the lowest total phenol content of 0.98 mg/g. The total phenolic content of alfalfa sprouts in the 2R2B, 1R3B, and blue light treatments were not significantly different from the total phenol content of the other treatments. Zhang et al. showed that the expression of *SmPAL1* and *Sm4CL1* was significantly increased by irradiation with mixed red and blue color light. Both of these two genes, as well as the promoters of the genes, respond to red-blue light, leading to the highest levels of phenolic compounds in *Salvia miltiorrhiza* Bunge when exposed to red–blue light [43]. The results of this experimental study showed that blue light treatment favored the accumulation of total phenolic content in alfalfa sprouts, which was similar to the previous study. Blue light could significantly increase the content of all phenols, thus elevating the total phenolic content. The total phenolic content of alfalfa sprouts illuminated with a mixture of red and blue light was high compared to blue light alone, perhaps because the mixed-color light benefited the expression of some of the phenolic acids basically.

### 3.7. Effect of Different Light-Quality Treatments on the Total Antioxidant Activity of Alfalfa Sprouts

DPPH is commonly used in chemical reactions involving free radicals to reflect the total antioxidant level of a substance [28,44]. The higher the DPPH free radical-scavenging rate, the higher the total antioxidant capacity of the substance. Figure 6 shows the effect of different light qualities on DPPH radical-scavenging rate in alfalfa sprouts. The blue light treatment had the highest DPPH radical-scavenging rate of 55.86%. It was followed by 1R3B treatment and 2R2B treatment, with 52.07% and 51.50%, respectively. Alfalfa sprouts in the dark treatment had the worst effect on the DPPH radical-scavenging rate of 47.80%. Plant antioxidant activity can be enhanced not only by phenolic acids, but also by quercetin, flavonoids, anthocyanins, vitamin C, photosynthetic pigments antioxidant

enzymes and other substances [45]. Blue light might promote the production of these antioxidants, elevating the overall antioxidant level in the plant bodies.

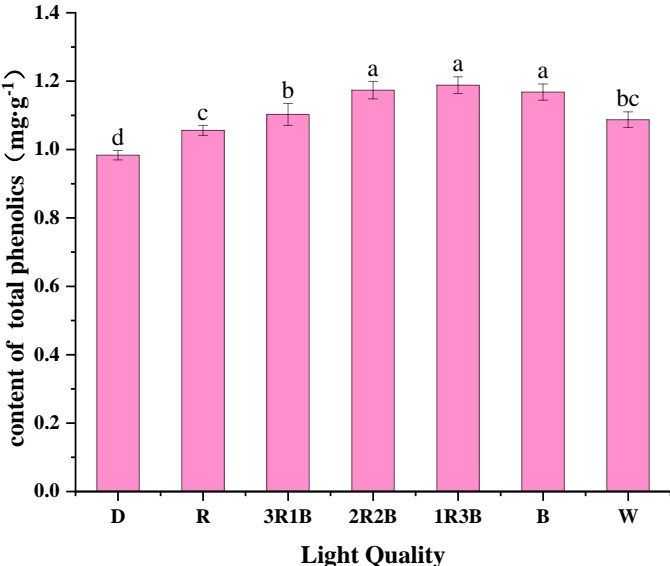

**Figure 5.** Effect of different light qualities on the total phenolic content of alfalfa sprouts.

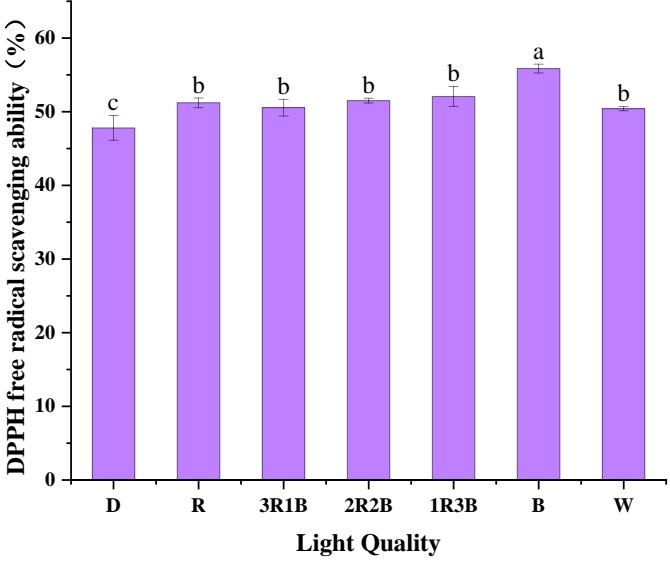

**Figure 6.** Free radical scavenging ability of DPPH on alfalfa sprouts under the influence of different light qualities.

### 4. Conclusions

In this experiment, alfalfa sprouts with a sprout length of 3 cm were treated with LEDs of different light qualities under $25 \pm 2$ °C, a light intensity of 30 μmol·m$^{-2}$·s$^{-1}$, and a light photoperiod of 12 h for 3 days. The experimental results showed that growth and antioxidant properties of alfalfa sprouts are differently affected by different light qualities of illumination. The dark environment favored the growth of hypocotyls and the increase in soluble sugar content of alfalfa sprouts; red light treatment increased the fresh weight and water content of alfalfa sprouts; different red and blue light treatments had little effect on the dry weight of alfalfa sprouts; the 2R2B treatment had the lowest nitrate content; the 1R3B treatment had the highest content of soluble proteins and total phenols; and the blue light-treated alfalfa sprouts had the best overall antioxidant performance. Considering the

nutritional composition, food safety and antioxidant capacity, the 1R3B treatment is more suitable for the factory cultivation of alfalfa sprouts.

**Author Contributions:** K.S.: primary data processing, manuscript writing, experimental design; Y.P.: raw data processing, experimental assistance and image organization; M.W.: experimental assistance and related literature collection; W.L.: review and editing; Y.L.: review and editing; J.C.: funding acquisition, supervision. All authors have read and agreed to the published version of the manuscript.

**Funding:** This work was supported by the Fundamental Research Funds for the Central Universities (Program No. BC2023111).

**Data Availability Statement:** Raw and analyzed data as well as images from the experiments can be made available from the corresponding author.

**Acknowledgments:** We are grateful for the support Fundamental Research Funds of the Central Universities.

**Conflicts of Interest:** The authors declare no conflict of interest.

## Abbreviations

LED—light-emitting-diode; BSA—Bovine Serum Albumin; ABTS-2,2′—Azinobis-(3-ethylbenzthiazoline-6-sulphonate); FRAP—Ferric ion reducing antioxidant power; DPPH—1,1-Diphenyl-2-picrylhydrazyl radical.

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
