# Peer review of "Effect of Red and Blue Light on the Growth and Antioxidant Activity of Alfalfa Sprouts"

_horticulturae, doi:10.3390/horticulturae10010076_

Round 1

Reviewer 1 Report

Comments and Suggestions for Authors

The title of the manuscript narrows down the 'message', my suggestion is "....on the physiology of alfalfa sprouts"

Line74 At least 1 time you have to insert the latin name of alfalfa.

Line80 What do you mean on "cool"?

Line83 The applied light intensity is too low to get healthy sprouts, that is the reason, that you did not have significanz differences between the dry weights (Table 1)!

Line88 Missing the number of independent repetitions..

Line89 Not clear the number of the independent repetition per treatment (30? or 50?), please clarify it.

Line95 The same problem as before: what do you mean on "treatment group"?

Line102 What do you mean on "groups"? Need the number of independent replications per treatment.

Line103 This parameter is not equal with "water moister content', The water moister content is fresh weight-dry weigh in 'g' or 'mg', but this is a percentage, am I righ? Please chechk the name of this parameter!

Line105 Need to correct the name of this sub-chapter, my suggestion is: biochemical parameters instead of physiological.

Line151 The replication number is here for "experimental data", but still not clear the number of independent repetitions of treatments. Did you accept significance at p=0.05?  What is your control?

Line155 The Figure1 alone is not enough, need data! The title of Figure 1 is too short, need to explain what we can see! When was it taken? Where is the baseline? Need a line between sooh and root, plus need ruler as 'Y' axe to help to catch the differences. I found it later in the next sub-chapter, but have to link it here.

Line163 "quite large' is not scientific, what do you mean on?

Line159 The title if Table1 is too short, need to explain what is in the table. What does the small letter meaning? Delete the "10 plants" from the table, you already declaired it in MAterial and Methods chapter.

Line170 Did you accept significance at p=0.05? You mean p≤0.05?

Line171 Please leave the "group"! Shade treatment, anyway, it was shade or dark????

Line202 Figure 2, What does the letters mean? "D"...? Need to write a detailed title for this figure as well. Hard to belive, that you have a significant differences based on you figures.

Author Response

Thank you for your comments, please see attachment.

Reviewer 2 Report

Comments and Suggestions for Authors

The authors used different ratios of red and blue LEDs as light sources for alfalfa sprout growth. Bimetric indicators and antioxidant properties were studied.

I have a few comments:

1. The abstract should be expanded by adding conditions for growing alfalfa sprouts.

2. Indicate the brand of equipment used for measurements.

3. In paragraph 2.1, indicate the wavelength of blue and red light. Add white light settings as well (spectrum, color temperature, etc.). How was the ratio of red and blue light calculated? Is it the ratio of the PPPD or the number of LEDs? Why was such low irradiance chosen?

4. In the caption to Figures 2-6, indicate the definition of the letter and other symbols.

5. Figure 5 without caption.

6. Section 3 should be divided into subsections.

7. The discussion should compare the results obtained with other similar works, for example https://doi.org/10.5333/KGFS.2023.43.3.177, 10.3390/horticulturae8030217

Author Response

Thank you for your comments, please see the attachment.

Reviewer 3 Report

Comments and Suggestions for Authors

Given that the sugar content is highest in plants grown in the dark, protein in plants grown in a combination of red and blue light, phenol content in plants grown in blue light, but also in a combination of red and blue, and the antioxidant activity is highest in plants grown in blue light, maybe the conclusion to recommend production alfalfa sprouts in 1R3B is not entirely accurate.

Specific comments:

Line 24: "Soybean sprouts" is repeated. Please delete one.

Line 78 There are no alfalfa sprout seeds. These are alfalfa seeds that will become sprouts after sprouting.

Lines 122-128: Match the tense of the verb with the rest of the text in paper (weighed, added…)

Lines 120, 128, 136: Remove the space between the last word in the sentence and the period

Lines 138-146: Match the tense of the verb with the rest of the text in paper

Line 140: The expression "try liquid nitrogen-assisted grinding" is not clear, why "try"

Line 177: Please explain what is "Lorena experimental results"

Lines 215 and 216: Are you referring to the sugar content? If to add, if not, to explain what these expressions mean - "The content of sprouts" and "the content of alfalfa sprouts".

Line 218: Is the term "shoots" correct in this case?

Line 250: What does the number 2 after the period at the end of the sentence represent?

Line 271: State the meaning of the abbreviations ABTS, FRAP

Lines 277 and 304: Is it shade or dark? It's not the same.

Lines 280 and 281: Please add before SmPAL1 and Sm4CL1 that these are genes and put them in italics.

Line 283: Salvia miltiorrhiza put in italic

Line 291: Not a good title - replace phenolic content with antioxidant activity

Lines 304 – 305: Rewrite the sentence so that the same expressions "radical scavenging rate" are not repeated

Lines 311-313: This sentence is not needed here.

One gets the impression that the paper was not proofread before the application, and hence there are a lot of technical errors.

Author Response

(The authors gave the same response as above.)

Round 2

Reviewer 2 Report

Comments and Suggestions for Authors

The authors have corrected the manuscript. I have no more questions.